# Genetic Polymorphisms Associated with Perioperative Joint Infection following Total Joint Arthroplasty: A Systematic Review and Meta-Analysis

**DOI:** 10.3390/antibiotics11091187

**Published:** 2022-09-02

**Authors:** Adel Hijazi, Ahmad Hasan, Adam Pearl, Ramiz Memon, Michael Debeau, Mariana Roldan, Mohamed E. Awad, Ehsen Abdul-Kabir, Khaled J. Saleh

**Affiliations:** 1School of Medicine, Wayne State University, Detroit, MI 48201, USA; 2Department of Orthopaedic Surgery, Detroit Medical Center, Detroit, MI 48201, USA; 3Department of Orthopaedic Surgery, John D Dingell VAMC, Detroit, MI 48201, USA; 4School of Medicine, Oakland University William Beaumont, Rochester, MI 48309, USA; 5School of Medicine, Universidad Autónoma de Guadalajara, Zapopan 45129, Mexico; 6College of Human Medicine, Michigan State University, Lansing, MI 48824, USA; 7School of Medicine, Central Michigan University, Mount Pleasant, MI 48859, USA

**Keywords:** total joint, infection, PJI, genetic, polymorphism, arthroplasty, hip, knee, association, nucleotide

## Abstract

The number of orthopedic procedures, especially prosthesis implantation, continues to increase annually, making it imperative to understand the risks of perioperative complications. These risks include a variety of patient-specific factors, including genetic profiles. This review assessed the current literature for associations between patient-specific genetic risk factors and perioperative infection. The PRISMA guidelines were used to conduct a literature review using the PubMed and Cochrane databases. Following title and abstract review and full-text screening, eight articles remained to be reviewed—all of which compared single nucleotide polymorphisms (SNPs) to periprosthetic joint infection (PJI) in total joint arthroplasty (TJA). The following cytokine-related genes were found to have polymorphisms associated with PJI: TNFα (*p* < 0.006), IL-6 (*p* < 0.035), GCSF3R (*p* < 0.02), IL-1 RN-VNTR (*p* = 0.002), and IL-1B (*p* = 0.037). Protein- and enzyme-related genes that were found to be associated with PJI included: MBL (*p* < 0.01, *p* < 0.05) and MBL2 (*p* < 0.01, *p* < 0.016). The only receptor-related gene found to be associated with PJI was VDR (*p* < 0.007, *p* < 0.028). This review compiled a variety of genetic polymorphisms that were associated with periprosthetic joint infections. However, the power of these studies is low. More research must be conducted to further understand the genetic risk factors for this serious outcome.

## 1. Introduction

Orthopedic surgeries, such as ACL repair and primary total arthroplasties, register among the most effective means to treat musculoskeletal disease and injury. The incidence of these orthopedic procedures has grown with projections demonstrating increased demand in the coming years [1,2,3]. In order to maximize the therapeutic benefit of these procedures and decrease patient costs, it is imperative to minimize associated perioperative complications [4,5]. Perioperative infection, particularly periprosthetic joint infection (PJI), represents a serious complication that decreases the therapeutic benefit of arthroplasty and increases morbidity and mortality.

Periprosthetic joint infection (PJI) is associated with longer hospital stays and increased costs, primarily due to the preferred course of treatment: revision surgery [6,7]. With an incidence of 1.2% and 4.6% for THA and TKA, respectively, decreasing occurrence requires a major effort among researchers and clinicians. Treating PJI-related arthroplasty failure with revision surgery is further associated with increased risk of infection, blood loss, surgical time, and complications, which adds additional costs to treatment [8,9,10,11]. Following infection, the innate immune system is activated to produce inflammatory mediators, such as interleukins and TNF-α, to mediate the immune effector response [12]. The genes for several of these proteins exhibit variability between individuals that may alter susceptibility to PJI [13,14,15]. Mannose binding lectin (MBL) is an acute phase protein that participates in innate immunity by opsonizing bacteria and activating complement [16]. Matrix metalloproteinase 1 (MMP-1) is a protease that breaks down interstitial collagens and cytokines that mediate the innate immune response [12,17]. There are a variety of protein receptors whose function may alter susceptibility to PJI. Toll-like Receptors 2 and 4 (TLR2, TLR4) are important in recognizing pathogen-associated molecular pattern (PAMP) molecules and mediating the innate immune response [18]. The RANK/RANKL/OPG pathway has been implicated in both bone homeostasis and immunity, revealing a possible role in PJI genetic risk assessment [19].

A genetic polymorphism is a variation in the genetic code of a particular gene that occurs in more than 1% of the population. Genetic variants include single nucleotide polymorphisms (SNPs), frame-shift mutations, chromosomal abnormalities, translocations, deletions, and duplications. Many genetic polymorphisms associated with PJI following TJA have been identified, particularly those related to antithrombin, protein C, protein S, factor V Leiden mutation, and thrombin [13,14,15,20,21,22,23,24]. The genes in these studies primarily encoded proteins that are involved in innate immunity. This systematic review was conducted to identify and compile research on possible associations between patient genetic risk factors and perioperative infection. Identified risk factors may serve as additional prophylactic criteria to decrease poor outcomes and costs, as well as augment informed consent during patient counseling.

## 2. Materials and Methods

### 2.1. Protocol

This systematic review and meta-analysis were conducted according to the preferred reporting items for systematic review and meta-analysis (PRISMA) guidelines and the Cochrane handbook.

### 2.2. Literature Search

A comprehensive literature search was performed using the PubMed and Cochrane databases. To ensure inclusion of all available evidence, we also manually searched the references of previous studies for other studies which met our inclusion criteria. The search strategy was a combination of subject headings and free text words, and the following MeSH term keywords were used: genetic polymorphism; implantation, joint prosthesis, total hip arthroplasty, complication, joint revision, orthopedic, postoperative, genome wide association study, and genetic susceptibility.

### 2.3. Eligibility Criteria

Inclusion and exclusion criteria were derived from population, intervention, comparison, and outcome (PICO) and non-PICO based exclusion taxonomy. Inclusion criteria were defined as any publication measuring association or causation between patient genetic risk factors and periprosthetic infection in any orthopedic surgery. Genetic risk factors included single nucleotide polymorphisms (SNPs), frame-shift mutations, chromosomal abnormalities, translocations, deletions, and duplications. Orthopedic surgeries included total shoulder (TSA), total knee (TKA), and total hip arthroplasties (THA). Exclusion criteria included comparisons of any non-genetic risk factors to perioperative infection, or comparisons of genetic risk factors to any non-infection perioperative complication. There were no exclusions based on study location, surgical approach, type of material used for implant fixation, prosthesis type, or other differences in surgical procedures.

### 2.4. Study Selection

Two reviewers (AH, AIH) independently screened abstracts and full-text articles derived by the search strategy and selected eligible articles based on the inclusion criteria. Reviewers then jointly addressed differences in the outcomes of the independent screenings.

### 2.5. Characteristics of Included Studies

Two independent reviewers (AH, MR) extracted information from all included articles. A data collection spreadsheet was created to sort quantitative and qualitative data for analysis. The data were extracted using the following variables: (1) demographics and characteristics (author, country of trial, year of publication, sample size, sample ethnicity mean age, and BMI), (2) type of surgery, (3) thromboprophylaxis, and (4) genetic polymorphism and gene detection. In addition, primary outcome variable (incidence of PJI events) was also extracted.

### 2.6. Strengthening the Reporting of Genetic Association Studies (STREGA)

Two reviewers (AH, MR) independently applied an 11-item quality checklist, derived from the STREGA (Strengthening the Reporting of Genetic Association Studies) and STROBE (Strengthening the Reporting of Observational Studies in Epidemiology) checklists to evaluate the methodological quality of the included studies. This 11-item quality checklist includes: (1) clear statement of objectives and hypothesis, (2) clear eligibility criteria for study participants, (3) clear definition of all variables, (4) clear diagnostic criteria, (5) replicability of statistical methods, (6) credible genetic testing method, (7) assessment of Hardy–Weinberg equilibrium, (8) assessment of ethnicity, and addressing the problem of mixed ethnicities statistically (if applicable), (9) sufficient descriptive data (age, sex, ethnicity), (10) statement of genotype frequencies, and (11) consideration of population stratification. Statistical significance was represented by *p*-values less than 0.05. Characteristics of the studies were compiled and reported.

### 2.7. GeneMANIA: Genetic Pathways and Interactions of PJI

GeneMANIA (http://www.genemania.org, accessed on 1 August 2022) provides a flexible, user-friendly analytics web interface for generating hypotheses based on gene functions, analyzing gene lists and prioritizing genes for functional assays. We adopted GeneMANIA to construct a gene–gene interaction network for genetic interaction involved in the incidence of PJI in terms of physical interactions, co-expression, predictions, co-localization, and genetic interaction, as well as to evaluate their functions.

### 2.8. Meta-Analysis

Standard meta-analytic methods were used to combine the results of all studies that provided sufficient data to obtain overall effect size estimates and the corresponding forest plots. Cochran’s Q statistic was used to assess heterogeneity of the studies, and publication bias was assessed using funnel plots and fail-save analyses. I2 values were calculated to estimate the heterogeneity among the included studies. In the presence of homogeneity (I2 < 50%), the fixed effects model was used to estimate the overall effects. If there was significant heterogeneity among included studies, the random effects model was used. The meta-analysis was performed using RevMan 5.3 software.

## 3. Results

### 3.1. Literature Search

A total of 299 manuscripts from 1997 to 2019 were retrieved via Pubmed (287) and Cochrane (12) databases during the initial identification phase (Figure 1). Of these, 155 manuscripts remained after removing duplicates. These manuscripts were then screened using inclusion and exclusion criteria via title and abstract review. The remaining 10 manuscripts underwent a full text screen for the inclusion and exclusion criteria, after which 8 remained for inclusion in our study. One study examined PJIs but was excluded due to not addressing SNPs. The other excluded study mentioned osteolysis in the title, but made no mention of PJI.

### 3.2. Study Characteristics

Of the eight included studies, seven are case-control studies and one is a prospective cohort study. Publication years ranged from 2006 to 2018, and all studies evaluated the association between PJI and SNP expression, of which 27 SNPs were reviewed. A total of 2308 patients were enrolled in these eight studies. Fifty-nine revision TKAs, 29 revision THAs, 1916 primary THAs, and 367 primary TKAs were included. The overwhelming majority of studies (7/8) were conducted with Caucasian populations, while one study was conducted with a Turkish population. The mean age at surgery for PJI and control patients across all studies was 66 and 58 years of age, respectively. Included studies were published in the following countries: Czech Republic (4/8), United Kingdom (3/8), and Turkey (1/8), as seen in Table 1.

### 3.3. Quality Assessment of Studies

Based on the 11-item quality checklist derived from STREGA and STROBE, the studies included in this meta-analysis contained an average score of 10.25, with a median of 11, mode of 11, and range of 7 to 11. Most of the studies fulfilled all the requirements to generate a strong quality assessment score, including stating the objectives and hypothesis, providing clear eligibility criteria, defining all variables, containing replicable statistical methods, providing sufficient descriptive data (e.g., age, gender), and stating genotype frequencies. The descriptive statistics for each criterion of the STREGA and STROBE methodology score are shown in Table 2.

### 3.4. Interaction and Genetic Pathways of Prosthetic Joint Infection (GeneMANIA)

The gene-gene interaction network generated by GeneMANIA identified seven genes linked to PJI: mannose binding lectin 2 (MBL2), colony stimulating factor 3 (CSF3), interleukin receptor 1 antagonist (IL1RN), vitamin D receptor (VDR), tumor necrosis factor (TNF), interleukin 6 (IL6), and interleukin 1B (IL1B). An additional 20 peripherally linked genes were also identified and can be found in Figure 2. There were five types of gene–gene network connections identified, including: co-expression, shared protein domains, co-localization, pathway, and genetic interactions.

### 3.5. Systematic Review

Of the eight studies, minimal overlap existed among the specific genes analyzed across all studies. Therefore, only statistically significant findings for genes analyzed once are reported below.

#### 3.5.1. Cytokines

Four studies reviewed twelve genes encoding for cytokines and cytokine receptors (IL-1, IL-1B, IL-4, IL-6, IL-8, IL-12A, IL-12B, IL-17A, IL-17F, IL-23R, TNF-α, GM-CSF) in association with the development of PJI [14,20,22]. Erdemli et al., in a prospective study of 88 patients undergoing revision arthroplasty, found significant associations between the A allele at position 238 in TNFα (*p* < 0.0006), the 1/2 and 2/2 alleles in IL-1 RN-VNTR, IL-6, and CT genotype in GCSF3R (*p* < 0.02) with PJI, but no association between IL-8 and IL-17 with PJI [20]. Stahelova et al. evaluated SNPs in the genes for IL-1B, TNF, and IL-6 for associations with PJI via a case control study of 303 patients who underwent TJA and 168 healthy Czech individuals who did not receive a TJA [22]. Among the TJA group, 89 patients developed PJI while the remaining 214 did not. They found that increased T allele frequency at position 511 in the gene for IL-1B increased risk of PJI development in the TJA group (pcorr = 0.037). No other significant associations were found [22].

#### 3.5.2. Proteins and Enzymes

Malik et al., in a case-control study of 162 revision THAs and 150 controls, found significant associations between MBL SNPs and PJI [13]. In the THA group, 71 experienced septic failure and the remaining 91 had aseptic failure. They found that the C allele (*p* < 0.01) and C/C genotype (*p* < 0.05) frequency at position 550 increased risk of PJI. G/G genotype frequency at Codon 54 SNP also increased risk (*p* < 0.05). No significant associations between codon 52 SNP or promoter 221 SNP and PJI were discovered [13]. Navratilova et al., in a case-control study of 357 revision TJAs (112 septic, 245 aseptic) and 196 controls, found that the L allele of MBL2 (550 SNP) was significantly more frequent in the septic group than the non-TJA control group (*p* < 0.010) and septic group (*p* < 0.016) [15].

Malik et al., in a case-control study of 162 THA’s (71 septic failure, 91 aseptic failure) and 150 controls, found no association between MMP1-1, MMP1-3, or MMP1-4 and PJI [14].

#### 3.5.3. Receptors

Mrazek et al., in a case-control study of 350 TJA patients (98 septic failure, 252 aseptic failure) and 189 non-TJA controls, found no differences in TLR2 or TLR4 allele frequencies between the PJI group and both the aseptic failure and control group [23]. Vitamin D Receptors (VDR) are important in bone-remodeling pathways, however their relevance to PJI has not been well studied [25]. Malik et. al, found that VDR SNPs were associated with osteolysis due to deep infection; specifically, the T allele (*p* < 0.007) and T/T genotype (*p* < 0.028) [14].

#### 3.5.4. RANK/RANKL/OPG Pathway

Malik et al., in a case-control study of 162 THAs (71 septic failure, 91 aseptic failure) and 150 controls, found no significant association of SNPs in OPG-163, OPG-245, OPG+1181, or RANK+575 and PJI when compared to aseptic controls [21]. In agreement, Navratilova et al., in a case-control study of 348 TJAs (98 septic failure, 251 aseptic failure) and 185 population controls, found no significant association between OPG-163 SNPs and PJI [24].

### 3.6. Meta-Analysis

The C allele and C/C genotype of IL6-174 were the only polymorphisms applicable to meta-analysis, as they were the only polymorphisms studied across multiple studies.

Three studies examined the association of the C/C genotype of IL6-174 and PJI following TJA. Erdemli et al. reported significant associations between the C allele at position 174 in IL-6 (*p* < 0.035) and PJI [20]. However, both Malik et al. and Stahelova et al. found no association between the C allele IL-6 polymorphisms at the 174 position and septic failure [14,22]. Pooled results demonstrated no statistically significant association between C/C genotype polymorphisms of the IL6-174 gene and the incidence of PJI following TJA (OR= 0.90, 95% CI= 0.56–1.45, *p* = 0.67), as shown in Figure 3.

Two studies examined the association of the C allele of IL6-174 and PJI following TJA. Both Malik et al. and Stahelova et al. report no significant association between the C allele polymorphism and the incidence of PJI following TJA [14,22]. Our pooled results confirm this finding (OR= 1.13, 95% CI= 0.74–1.72; *p*= 0.56), as shown in Figure 4.

## 4. Discussion

Perioperative infection, particularly periprosthetic joint infection, is a major cause of prosthesis failure, morbidity, and mortality in patients undergoing orthopedic procedures. There are several biological and environmental risk factors that predispose patients to perioperative infection including: BMI, ASA scores, diabetes mellitus, alcohol use, and heart disease [26]. In addition, there has been identification of genetic differences among patients that may contribute to variability in risk for PJI. This review, in an attempt to identify any genetic risk factors for perioperative infection, was only able to find SNPs that are associated with infection in hip and knee arthroplasties. The significant genetic SNPs included TNFα-238 A allele, IL6-174 C allele, GCSF3R CT genotype, IL1 RN-VNTR 1/2 and 2/2 alleles, IL1B-511 T allele, MBL-550 C allele and C/C genotype, MBL codon 54 G/G genotype, MBL2-550 L allele, VDR T allele and T/T genotype.

Variability in genes that encode cytokine mediators may affect their transcription levels and consequently the innate immune response, therefore altering susceptibility to PJI [22]. This review identified TNFα-238 A allele, IL6-174 C allele, GCSF3R CT genotype, IL1 RN-VNTR 1/2 and 2/2 alleles, and the IL1B-511 T allele as increasing the risk of perioperative joint infection. However, it is important to note that while Erdemli et al. found SNPs at IL6-174 to be significant, Malik et al. and Stahelova et al. found no such significance [14,20,22]. It should be noted that Malik and Stahelova et al. both conducted case control studies while Erdemli et al. conducted a prospective study. All three studies had small patient populations and only utilized a single research center, decreasing the generalizability of the results.

Similarly, all the associations found between other significant genes and PJI had small, primarily Caucasian patient populations, which made conclusions less generalizable to the American population. Moreover, there were less than two publications examining any single SNP. In addition, data from these studies were unavailable for further statistical analyses that may have expanded upon the significance of identified associations.

This review did have limitations. There was publication bias, as it did not include articles written in languages other than English. Additionally, although a meta-analysis was conducted, this was performed using allele frequencies in septic vs. aseptic groups. Original data from the study authors were not extracted. Seven of the eight studies also assessed primary TJA, while only one assessed revision TJA. Theoretically, the proportion of individuals with these polymorphisms would be higher in the revision TJA group. Therefore, if a patient has septic loosening, screening for these polymorphisms could potentially have higher implications. This should be considered in future studies.

Advances in genetics have led to more than 4400 genes being identified as disease-related, with this number steadily increasing [27]. This has led to the commonality of genetic testing across specialties, from hypercholesterolemia and cardiac disease to genetic immunocompromising conditions. However, despite this data, there have been few guidelines put forth and very little data regarding the role of genetics in the development of infection. A major cause of this is that physicians test for particular genes, only doing so after specific concerns arise. It would be more beneficial and cost effective for patients to have full-genome and variant mapping. This can be as inexpensive as 300 USD and would provide clinicians with a host of information that can be cross-referenced with variant/mutation databases, helping to dictate diagnosis and treatment [28]. Such databases date as far back as 2005, when attempts were made to create a database of genomics that could lead to more tailored patient care. The HuGE Net, for example, sought to promote global efforts at a knowledge base to guide public health and disease prevention [29]. This network is still available, with global, North American, and European genomic databases. With the recent advances in artificial intelligence (AI) and machine-based learning algorithms, low cost and availability of genetic testing, and steps towards patient-specific care, similar databases should be considered for orthopedic infections. This necessitates a more robust data set, so that surgeon data can be combined with AI discerned patterns and prophylactic protocols for those with genetic polymorphisms predisposing them to PJI. It is not too far in the foreseeable future that the standard physical exam, blood work, and pre-operative radiographs are combined with genetic databases and AI protocols to help predict outcomes, complications, and resource utilization.

## 5. Conclusions

There are several SNPs in genes important to innate immunity that increase the risk of periprosthetic joint infection. TNFα-238 A allele, GCSF3R CT genotype, IL1 RN-VNTR 1/2 and 2/2 alleles, IL1B-511 T allele, and MBL C allele demonstrate increased risk of perioperative joint infection. Future studies should re-examine these SNPs to build a more robust body of evidence on their possible detrimental effects. These studies should have larger, more diverse populations studied in a prospective manner. Additional effort must be taken to share data between researchers for further statistical analyses. Phenomenal strides have been made in the field of genomics, becoming more intricate while more readily available and inexpensive. This provides a synergistic opportunity to create biobanks that utilize artificial intelligence to drive in a new era of individualized, patient-specific care.

## Figures and Tables

**Figure 1 antibiotics-11-01187-f001:**
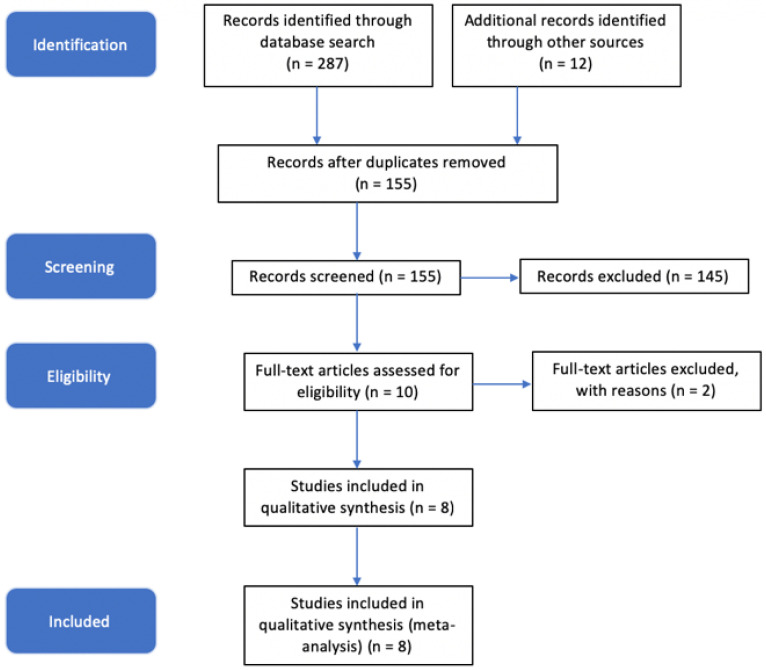
Results of systematic review-based literature search.

**Figure 2 antibiotics-11-01187-f002:**
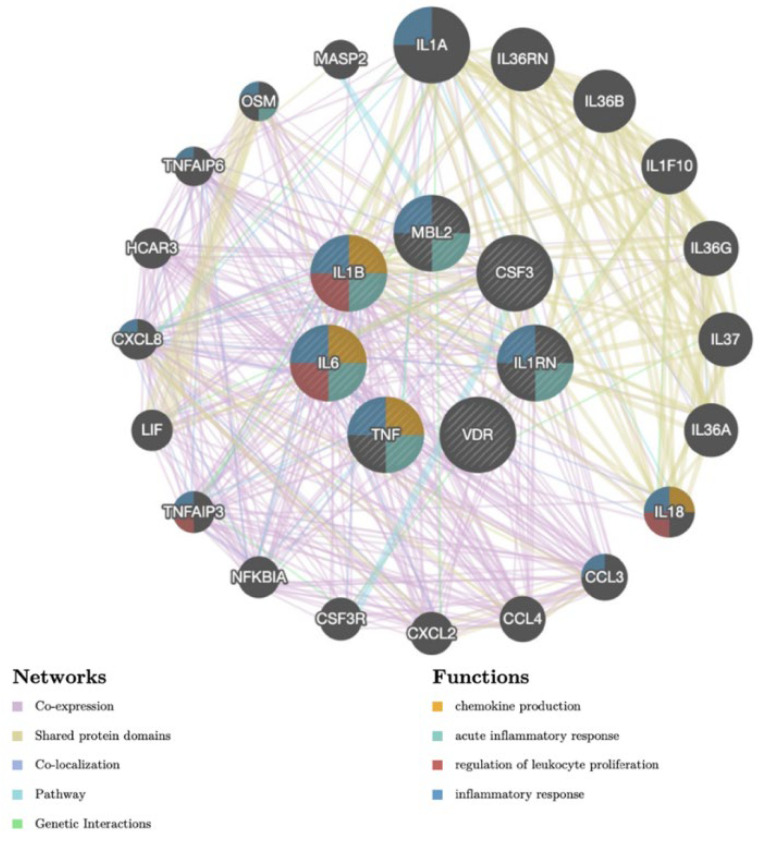
Gene-gene interaction network identifying seven genes linked to PJI.

**Figure 3 antibiotics-11-01187-f003:**
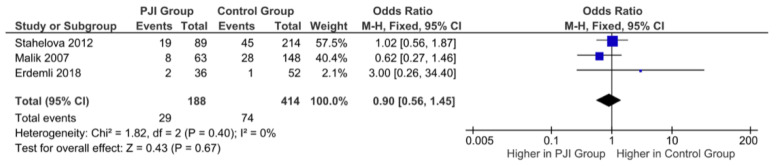
Forest plot indicating no statistically significant association between C/C genotype polymorphisms of the IL6-174 gene and the incidence of PJI following TJA.

**Figure 4 antibiotics-11-01187-f004:**
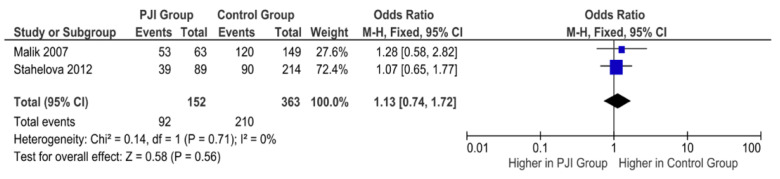
Forest plot indicating no significant association between the C allele polymorphism and the incidence of PJI following TJA.

**Table 1 antibiotics-11-01187-t001:** Characteristics of included studies.

Author (Year)	Genetic Polymorphism Studied	Study Type	Study Country	Sample Ethnicity	Sample Size	Mean Age at Surgery (All Groups)	Operation
Erdemli et al. (2018) [20]	TNFα-238 (A) IL6-174 (C) GCSF3R (T) IL1 RN-VNTR (1/2 and 2/2) IL1B-511 (T)	Prospective	Turkey	Turkish	88 (36 septic, 52 aseptic)	68	Revision arthroplasty
Malik et al. (2007) [13]	MBL-221 (promoter) MBL-550 (C) MBL Codon 54 (G) MBL Codon 52	Case-control	United Kingdom	White patients from Northwest England	312 (71 septic, 91 aseptic, 150 controls)	68.6	THA
Malik et al. (2006) [21]	OPG-163 OPG-245 OPG + 1181 RANK + 575	Case-control	United Kingdom	White patients from Northwest England	312 (71 septic, 91 aseptic, 150 non-TJA controls)	68.6	THA
Malik et al. (2007) b [14]	IL6-174 (C) MMP1-1 (C) MMP1-3 MMP1-4 MMP2 MMP4 VDR (T) VDR (L)	Case-control	United Kingdom	White patients from Northwest England	312 (71 septic, 91 aseptic, 150 non-TJA controls)	68.6	THA
Mrazek et al. (2013) [23]	TLR2 R753Q TLR4 D299G TLR4 T399I	Case-control	Czech Republic	Czech	350 (98 septic, 252 aseptic, 189 non-TJA controls)	64 (septic), 49 (aseptic)	TJA
Navratilova et al. (2012) [15]	MBL2-550 (L) MBL2-221 (X) MBL2+54 (A)	Case-control	Czech Republic	Czech	553 (112 septic, 245 aseptic, 196 non-TJA controls)	63 (septic), 50 (Aseptic), 29 (Control)	TJA
Navratilova et al. (2014) [24]	OPG-163 (C)	Case-control	Czech Republic	Czech	534 (98 septic, 251 aseptic, 185 non-TJA controls)	67 (PJI), 49 (Aseptic TJA), 28 (Control group)	TJA
Stahelova et al. (2012) [22]	IL1B-511 (T) IL1B+3962 TNF-308 TNF-238 IL6-174 IL6nt565	Case-control	Czech Republic	Czech	471 89 (PJI), 214 (no PJI), 168 non-TJA controls	63 (PJI) 47 (Aseptic TJA) not reported for control group	TJA

TNF, tumor necrosis factor; IL, interleukin; GCSF3R, granulocyte colony stimulating factor; MBL, mannose binding lectin; OPG, osteoprotegerin; RANK, receptor activator of nuclear factor κ B; MMP, matrix metalloproteinases; VDR, vitamin D receptor; TLR, toll-like receptor; THA: total hip arthroplasty; TJA: total joint arthroplasty.

**Table 2 antibiotics-11-01187-t002:** 11-Item Quality Assessment checklist derived from STREGA/STOBE Reporting.

Study	Introduction	Methods						Results			Discussion
	1. Objectives and hypothesis clearly stated	2. Clear eligibility criteria for participants/studies	3. Clear definition of all variables	4. Statistical methods replicable	5. Assessment of HWE	6. Assessment of ethnicity	7. Mixed ethnicities addressed statistically	8. Sufficient descriptive data (age, gender)	9. Genotype frequencies stated	10. Sample in HWE	11. Consideration of population
Erdemli et al. (2018) [20]	+	+	+	+	+	-	-	+	-	+	-
Malik et al. (2007)—MMP1 [14]	+	+	+	+	+	+	+	+	+	+	+
Malik et al. (2006)—RANK [21]	+	+	+	+	+	+	+	+	+	+	+
Malik et al. (2007)—MBL [13]	+	+	+	+	+	+	+	+	+	+	+
Mraszek et al. (2013) [23]	+	+	+	+	+	+	+	+	+	+	+
Navratilova et al. (2012) [15]	+	+	+	+	+	+	+	+	+	+	+
Navratilova et al. (2014) [24]	+	-	+	+	+	+	+	+	+	+	+
Stahelova et al. (2012) [22]	+	+	+	+	+	+	+	+	+	+	-

## Data Availability

Gene interaction data is publicly available and can be found at http://www.genemania.org (accessed on 1 August 2022).

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
