# Peer review of "Genetic Polymorphisms Associated with Perioperative Joint Infection following Total Joint Arthroplasty: A Systematic Review and Meta-Analysis"

_antibiotics, 2022, doi:10.3390/antibiotics11091187_

Round 1

Reviewer 1 Report

It is acceptable

Author Response

No response is necessary. We appreciate your time and thoughtful review!

Reviewer 2 Report

Dear authors,

First, I would like to express sincere gratitude to get an opportunity to review your manuscript.

The effort of the author is appreciated. Interesting systematic review and meta-analysis. A well-structured manuscript. After assessing the manuscript, the following issues raised my concerns or represent suggestions that in my opinion could increase the quality of the manuscript: Please reassess the manuscript for formatting and typing mistakes. As a personal opinion please rearrange de structure of the manuscript as the Materials and Methods section. Of the manuscript should be before the results section. 

Author Response

1. The materials/methods section was moved to after the introduction.

2. The manuscript was spell-checked and formatted

We appreciate your time and consideration!

Reviewer 3 Report

The manuscript entitled "Genetic polymorphism associated with perioperative joint infection following total joint arthroplasty: a systematic review and meta-analysis" is well designed and well written. It provides important information regaring the genes involved in the PJI. Also, it updates the clinicians and researchers about current situation in this filed. I recommed for publication of the mansucript in the current form.

Author Response

No response is necessary.

We appreciate your thoughtful review and comments!

Round 2

Reviewer 2 Report

Dear authors,

The manuscript deserves to be published. 

Author Response

Thank you for your response!